# Symmetric Rank-One Quasi-Newton Methods for Deep Learning Using Cubic Regularization

## Abstract

Stochastic gradient descent and other first-order variants, such as Adam and AdaGrad, are commonly used in the field of deep learning due to their computational efficiency and low-storage memory requirements. However, these methods do not exploit curvature information. Consequently, iterates can converge to saddle points or poor local minima. On the other hand, Quasi-Newton methods compute Hessian approximations which exploit this information with a comparable computational budget. Quasi-Newton methods re-use previously computed iterates and gradients to compute a low-rank structured update. The most widely used quasi-Newton update is the L-BFGS, which guarantees a positive semi-definite Hessian approximation, making it suitable in a line search setting. However, the loss functions in DNNs are non-convex, where the Hessian is potentially non-positive definite. In this paper, we propose using a limited-memory symmetric rank-one quasi-Newton approach which allows for indefinite Hessian approximations, enabling directions of negative curvature to be exploited. Furthermore, we use a modified adaptive regularized cubics approach, which generates a sequence of cubic subproblems that have closed-form solutions with suitable regularization choices. We investigate the performance of our proposed method on autoencoders and feed-forward neural network models and compare our approach to state-of-the-art first-order adaptive stochastic methods as well as other quasi-Newton methods.

## 1 Introduction

Deep learning problems often involve training deep neural networks (DNN) by minimizing an empirical risk of estimation, given by

$$\underset{\Theta \in \mathbb{R}^n}{\text{minimize}} \ f(\Theta) = \frac{1}{N} \sum_{i=1}^{N} f_i(x_i, y_i; \Theta) \tag{1}$$

where $\Theta$ is the vector of weights and each $f_i$ is a scalar-valued loss function that depends on a vector of data inputs, $x_i \in \mathbb{R}^{n_1}$, and outputs, $y_i \in \mathbb{R}^{n_2}$. Here, $N$ corresponds to the cardinality of the data set $\mathcal{D} = \{x_i, y_i\}$. In this paper, we assume that $f$ is continuously differentiable.

Gradient and adaptive gradient methods are the most widely used methods for solving (1). In particular, stochastic gradient descent (SGD), despite its simplicity, performs well over a wide range of applications. However, in a sparse training data setting, SGD performs poorly due to limited training speed (Luo et al., 2019). To address this problem, *adaptive* methods such as AdaGrad (Duchi et al., 2011), AdaDelta (Zeiler, 2012), RMSProp (Hinton et al., 2012) and Adam (Kingma & Ba, 2014) have been proposed. These methods take the root mean square of the past gradients to influence the current step.

In contrast, Newton's method has the potential to exploit curvature information from the second-order derivative (Hessian) matrix (see e.g., Gould et al. (2000)). Generally, the iterates are defined by $\Theta_{k+1} = \Theta_k - \alpha_k \nabla^2 f(\Theta_k)^{-1} \nabla f(\Theta_k)$, where $\alpha_k > 0$ is a steplength defined by a linesearch criterion (Nocedal & Wright, 2006). In a DNN setting, the number of parameters ($n$) can be of the order of millions. Thus, full Hessians are rarely ever computed. Instead, Hessian-vector products and Hessian-free methods are used (see e.g., Martens et al. (2010), Ranganath et al. (2021)) which reduce the cost of storing the Hessian and inverting it.

Quasi-Newton methods compute Hessian approximations, $\mathbf{B}_k \approx \nabla^2 f(\Theta_k)$, that satisfy the *secant condition* given by $\mathbf{y}_{k-1} = \mathbf{B}_k \mathbf{s}_{k-1}$, where

$$\mathbf{s}_{k-1} = \Theta_k - \Theta_{k-1} \quad \text{and} \quad \mathbf{y}_{k-1} = \nabla f(\Theta_k) - \nabla f(\Theta_{k-1}).$$

The most commonly used quasi-Newton method, including in the realm of deep learning, is the limited-memory BFGS update, or L-BFGS (see e.g., Liu & Nocedal (1989)), where the Hessian approximation is given by

$$\mathbf{B}_k = \mathbf{B}_{k-1} + \frac{\mathbf{y}_{k-1}\mathbf{y}_{k-1}^\top}{\mathbf{y}_{k-1}^\top \mathbf{s}_{k-1}} - \frac{\mathbf{B}_{k-1}\mathbf{s}_{k-1}\mathbf{s}_{k-1}^\top \mathbf{B}_{k-1}^\top}{\mathbf{s}_{k-1}^\top \mathbf{B}_{k-1}\mathbf{s}_{k-1}}. \tag{2}$$

One advantage of using an L-BFGS update is that the Hessian approximation can be guaranteed to be positive definite. This is highly suitable in line-search settings because the update $\mathbf{s}_k$ is guaranteed to be a descent direction. This means there is some step length along this direction that results in a decrease in the objective function (see Nocedal & Wright (2006), Algorithm 6.1). Because the L-BFGS update is positive definite, it does not readily detect directions of negative curvature for avoiding saddle points. In contrast, the Symmetric Rank-One (SR1) quasi-Newton update is not guarateed to be positive definite and can result in *ascent* directions for line-search methods. However, in trust-region settings where indefinite Hessian approximations are an advantage because they can capture directions of negative curvature, the limited-memory SR1 (L-SR1) has been shown to outperform L-BFGS in DNNs for classification (see Erway et al. (2020)). We discuss this in more detail in Section 2 but in the context of adaptive regularization using cubics (see e.g., Nesterov & Polyak (2006)).

**Contributions.** The main contributions of this paper are as follows: (1) The use of the L-SR1 update to model potentially indefinite Hessians of the non-convex loss function; (2) The use of adaptive regularization using cubics (ARCs) approach as an alternative to line-search and trust-region optimization methods; (3) The use of a shape-changing norm to define the cubic regularization term, which allows us to compute the closed form solution to the cubic subproblem in the ARCs approach; (4) Convergence proof of the proposed ARCs approach with L-SR1 Hessian approximations. To the knowledge of the authors, **this is the first time** a quasi-Newton approach has been used in an adaptive regularized cubics setting.

## 2 Proposed approach

In this section, we describe our proposed approach by first discussing the L-SR1 update.

**Limited-memory symmetric rank-one updates.** Unlike the BFGS update (2), which is a rank-two update, the SR1 update is a rank-one update, which is given by

$$\mathbf{B}_{k+1} = \mathbf{B}_k + \frac{(\mathbf{y}_k - \mathbf{B}_k\mathbf{s}_k)(\mathbf{y}_k - \mathbf{B}_k\mathbf{s}_k)^\top}{\mathbf{s}_k^\top(\mathbf{y}_k - \mathbf{B}_k\mathbf{s}_k)} \tag{3}$$

(see Khalfan et al. (1993)). As previously mentioned, $\mathbf{B}_{k+1}$ in (3) is not guaranteed to be definite. However, it can be shown that the SR1 matrices can converge to the true Hessian (see Conn et al. (1991) for details). We note that the pair $(\mathbf{s}_k, \mathbf{y}_k)$ is accepted only when

$$|\mathbf{s}_k^\top(\mathbf{y}_k - \mathbf{B}_k\mathbf{s}_k)| > \varepsilon\|\mathbf{s}_k\|_2\|\mathbf{y}_k - \mathbf{B}_k\mathbf{s}_k\|_2, \tag{4}$$

for some constant $\varepsilon > 0$ (see Nocedal & Wright (2006), Sec. 6.2, for details). The SR1 update can be defined recursively as

$$\mathbf{B}_{k+1} = \mathbf{B}_0 + \sum_{j=0}^{k} \frac{(\mathbf{y}_j - \mathbf{B}_j\mathbf{s}_j)(\mathbf{y}_j - \mathbf{B}_j\mathbf{s}_j)^\top}{\mathbf{s}_j^\top(\mathbf{y}_j - \mathbf{B}_j\mathbf{s}_j)}. \tag{5}$$

In limited-memory settings, only the last $m \ll n$ pairs of $(\mathbf{s}_j, \mathbf{y}_j)$ are stored and used. For ease of presentation, here we choose $k < m$. We define

$$\mathbf{S}_k = [\,\mathbf{s}_0 \ \ \mathbf{s}_1 \ \ \cdots \ \ \mathbf{s}_{k-1}\,] \quad \text{and} \quad \mathbf{Y}_k = [\,\mathbf{y}_0 \ \ \mathbf{y}_1 \ \ \cdots \ \ \mathbf{y}_{k-1}\,].$$

Then $\mathbf{B}_k$ admits a compact representation of the form

$$\mathbf{B}_k \;=\; \mathbf{B}_0 + \begin{bmatrix} \boldsymbol{\Psi}_k \end{bmatrix} \begin{bmatrix} \mathbf{M}_k \end{bmatrix} \begin{bmatrix} \boldsymbol{\Psi}_k^\top \end{bmatrix}, \tag{6}$$

where $\boldsymbol{\Psi}_k = \mathbf{Y}_k - \mathbf{B}_0 \mathbf{S}_k$ and

$$\mathbf{M}_k = (\mathbf{D}_k + \mathbf{L}_k + \mathbf{L}_k^\top - \mathbf{S}_k^\top \mathbf{B}_0 \mathbf{S}_k)^{-1},$$

where $\mathbf{L}_k$ is the strictly lower triangular part, $\mathbf{V}_k$ is the strictly upper triangular part, and $\mathbf{D}_k$ is the diagonal part of $\mathbf{S}_k^\top \mathbf{Y}_k = \mathbf{L}_k + \mathbf{D}_k + \mathbf{V}_k$ (see Byrd et al. (1994) for further details).

Because of the compact representation of $\mathbf{B}_k$, its partial eigendecomposition can be computed (see Erway & Marcia (2015)). In particular, if we compute the QR decomposition of $\boldsymbol{\Psi}_k = \mathbf{QR}$ and the eigendecomposition $\mathbf{RMR}^\top = \mathbf{P}\hat{\boldsymbol{\Lambda}}_k \mathbf{P}^\top$, then we can write

$$\mathbf{B}_k = \mathbf{B}_0 + \mathbf{U}_\parallel \hat{\boldsymbol{\Lambda}}_k \mathbf{U}_\parallel^\top,$$

where $\mathbf{U}_\parallel = \mathbf{QP} \in \mathbb{R}^{n \times k}$ has orthonormal columns and $\hat{\boldsymbol{\Lambda}}_k \in \mathbb{R}^{k \times k}$ is a diagonal matrix. If $\mathbf{B}_0 = \delta_k \mathbf{I}$ (see e.g., Lemma 2.4 in Erway et al. (2020)), where $0 < \delta_k < \delta_{\max}$ is some scalar and $\mathbf{I}$ is the identity matrix, then we obtain the eigendecomposition

$$\mathbf{B}_k = \mathbf{U}_k \boldsymbol{\Lambda}_k \mathbf{U}_k^\top = \begin{bmatrix} \mathbf{U}_\parallel & \mathbf{U}_\perp \end{bmatrix} \begin{bmatrix} \hat{\boldsymbol{\Lambda}}_k + \delta_k \mathbf{I} & 0 \\ 0 & \delta_k \mathbf{I} \end{bmatrix} \begin{bmatrix} \mathbf{U}_\parallel^\top \\ \mathbf{U}_\perp^\top \end{bmatrix} \tag{7}$$

where $\mathbf{U}_k = \begin{bmatrix} \mathbf{U}_\parallel & \mathbf{U}_\perp \end{bmatrix}$ is an orthogonal matrix and $\mathbf{U}_\perp \in \mathbb{R}^{n \times (n-k)}$ is a matrix whose columns form an orthonormal basis orthogonal to the range space of $\mathbf{U}_\parallel$. Here,

$$(\boldsymbol{\Lambda}_k)_i = \begin{cases} \delta_k + \hat{\lambda}_i & \text{if } i \le k \\ \delta_k & \text{if } i > k \end{cases}. \tag{8}$$

**Adaptive regularization using cubics.** Since the SR1 Hessian approximation can be indefinite, some safeguard must be implemented to ensure that the resulting search direction $\mathbf{s}_k$ is a descent direction. One such safeguard is to use a "regularization" term. The Adaptive Regularization using Cubics (ARCs) method (see Griewank (1981); Nesterov & Polyak (2006); Cartis et al. (2011)) can be viewed as an alternative to line-search and trust-region methods. At each iteration, an approximate global minimizer of a local (cubic) model,

$$\min_{\mathbf{s} \in \mathbb{R}^n} m_k(\mathbf{s}) \equiv \mathbf{g}_k^\top \mathbf{s} + \frac{1}{2} \mathbf{s}^\top \mathbf{B}_k \mathbf{s} + \frac{\mu_k}{3} (\Phi_k(\mathbf{s}))^3, \tag{9}$$

is determined, where $\mathbf{g}_k = \nabla f(\Theta_k)$, $\mu_k > 0$ is a regularization parameter, and $\Phi_k$ is a function (norm) that regularizes $\mathbf{s}$. Typically, the Euclidean norm is used. In this work, we use an alternative "shape-changing" norm that allows us to solve each subproblem (9) exactly. Proposed in Burdakov et al. (2017), this shape-changing norm is based on the partial eigendecomposition of $\mathbf{B}_k$. Specifically, if $\mathbf{B}_k = \mathbf{U}_k \boldsymbol{\Lambda}_k \mathbf{U}_k^\top$ is the eigendecomposition of $\mathbf{B}_k$, then we can define the norm

$$\|\mathbf{s}\|_{\mathbf{U}_k} \overset{\text{def}}{=} \|\mathbf{U}_k^\top \mathbf{s}\|_3.$$

It can be shown using Hölder's Inequality that

$$\frac{1}{\sqrt[6]{n}} \|\mathbf{s}\|_2 \le \|\mathbf{s}\|_{\mathbf{U}_k} \le \|\mathbf{s}\|_2.$$

As per the authors' literature review, this is the first time the adaptive regularized cubics has been used in conjunction with a shape changing norm in a deep learning setting. The main motivation of using this adaptive regularized cubics comes from better convergence properties when compared with a trust-region approach (see Cartis et al. (2011)). Using the shape-changing norm allows us to solve the subproblem exactly.

**Closed-form solution.** Applying a change of basis with $\bar{\mathbf{s}} = \mathbf{U}_k^\top \mathbf{s}$ and $\bar{\mathbf{g}}_k = \mathbf{U}_k^\top \mathbf{g}_k$, we can redefine the cubic subproblem as

$$\min_{\bar{\mathbf{s}} \in \mathbb{R}^n} \bar{m}_k(\bar{\mathbf{s}}) = \bar{\mathbf{g}}_k^\top \bar{\mathbf{s}} + \frac{1}{2} \bar{\mathbf{s}}^\top \mathbf{\Lambda}_k \bar{\mathbf{s}} + \frac{\mu_k}{3} \|\bar{\mathbf{s}}\|_3^3. \tag{10}$$

With this change of basis, we can easily find a closed-form solution of (10), which is generally not the case for other choices of norms. Note that $\bar{m}_k(\bar{\mathbf{s}})$ is a separable function, meaning we can write $\bar{m}_k(\bar{\mathbf{s}})$ as

$$\bar{m}_k(\bar{\mathbf{s}}) = \sum_{i=1}^n \left\{ (\bar{\mathbf{g}}_k)_i (\bar{\mathbf{s}})_i + \frac{1}{2} (\mathbf{\Lambda}_k)_i (\bar{\mathbf{s}})_i^2 + \frac{\mu_k}{3} |(\bar{\mathbf{s}})_i|^3 \right\}.$$

Consequently, we can solve (10) by solving one-dimensional problems of the form

$$\min_{\bar{s} \in \mathbb{R}} \; \bar{m}(\bar{s}) = \bar{g}\bar{s} + \frac{1}{2}\lambda\bar{s}^2 + \frac{\mu_k}{3}|\bar{s}|^3, \tag{11}$$

where $\bar{g} \in \mathbb{R}$ corresponds to entries in $\bar{\mathbf{g}}_k$ and $\lambda \in \mathbb{R}$ corresponds to diagonal entries in $\mathbf{\Lambda}_k$. To find the minimizer of (11), we first write $\bar{m}(\bar{s})$ as follows:

$$\bar{m}(\bar{s}) = \begin{cases} \bar{m}_+(s) = \bar{g}\bar{s} + \frac{1}{2}\lambda\bar{s}^2 + \frac{\mu_k}{3}\bar{s}^3 & \text{if } \bar{s} \geq 0, \\ \bar{m}_-(\bar{s}) = \bar{g}\bar{s} + \frac{1}{2}\lambda\bar{s}^2 - \frac{\mu_k}{3}\bar{s}^3 & \text{if } \bar{s} \leq 0. \end{cases}$$

The minimizer $\bar{s}^*$ of $\bar{m}(\bar{s})$ is obtained by setting $\bar{m}'(\bar{s})$ to zero and will depend on the sign of $\bar{g}$ because $\bar{g}$ is the slope of $\bar{m}(\bar{s})$ at $\bar{s} = 0$, i.e., $\bar{m}'(0) = \bar{g}$. In particular, if $\bar{g} > 0$, then $\bar{s}^*$ is the minimizer of $\bar{m}_-(\bar{s})$, namely $\bar{s}^* = (-\lambda + \sqrt{\lambda^2 + 4\bar{g}\mu})/(-2\mu)$. If $\bar{g} < 0$, then $\bar{s}^*$ is the minimizer of $\bar{m}_+(\bar{s})$, which is given by $\bar{s}^* = (-\lambda + \sqrt{\lambda^2 - 4\bar{g}\mu})/(2\mu)$. Note that these two expressions for $\bar{s}^*$ are equivalent to the following formula:

$$\bar{s}^* = \frac{-2\bar{g}}{\lambda + \sqrt{\lambda^2 + 4|\bar{g}|\mu}},$$

In the original space, $\mathbf{s}^* = \mathbf{U}_k \bar{\mathbf{s}}^*$ and $\mathbf{g}_k = \mathbf{U}_k \bar{\mathbf{g}}_k$. Letting

$$\mathbf{C}_k = \text{diag}(\bar{c}_1, \ldots, \bar{c}_n), \quad \text{where} \;\; \bar{c}_i = \frac{2}{\lambda_i + \sqrt{\lambda_i^2 + 4|\bar{g}_i|\mu}}, \tag{12}$$

then the solution $\mathbf{s}^*$ in the original space is given by

$$\mathbf{s}^* = \mathbf{U}_k \bar{\mathbf{s}}^* = -\mathbf{U}_k \mathbf{C}_k \mathbf{U}_k^\top \mathbf{g}_k. \tag{13}$$

**Practical implementation.** While computing $\mathbf{U}_\| \in \mathbb{R}^{n \times k}$ in the matrix $\mathbf{U}_k = [\, \mathbf{U}_\| \;\; \mathbf{U}_\perp \,]$ is feasible since $k \ll n$, computing $\mathbf{U}_\perp$ explicitly is not. Thus, we must be able to compute $\mathbf{s}^*$ without needing $\mathbf{U}_\perp$. First, we define the following quantities

$$\bar{\mathbf{s}}_\| = \mathbf{U}_\|^\top \mathbf{s} \quad \text{and} \quad \bar{\mathbf{s}}_\perp = \mathbf{U}_\perp^\top \mathbf{s},$$
$$\bar{\mathbf{g}}_\| = \mathbf{U}_\|^\top \mathbf{g}_k \quad \text{and} \quad \bar{\mathbf{g}}_\perp = \mathbf{U}_\perp^\top \mathbf{g}_k.$$

Then the cubic subproblem (10) becomes

$$\underset{\bar{\mathbf{s}} \in \mathbb{R}^n}{\text{minimize}} \; \bar{m}_k(\bar{\mathbf{s}}) \;=\; \underset{\bar{\mathbf{s}}_\| \in \mathbb{R}^k}{\text{minimize}} \; \bar{m}_\|(\bar{\mathbf{s}}_\|) + \underset{\bar{\mathbf{s}}_\perp \in \mathbb{R}^{n-k}}{\text{minimize}} \; \bar{m}_\perp(\bar{\mathbf{s}}_\perp), \tag{14}$$

where

$$\bar{m}_\|(\bar{\mathbf{s}}_\|) \;=\; \bar{\mathbf{g}}_\|^\top \bar{\mathbf{s}}_\| + \frac{1}{2} \bar{\mathbf{s}}_\|^\top \hat{\mathbf{\Lambda}}_k \bar{\mathbf{s}}_\| + \frac{\mu_k}{3} \|\bar{\mathbf{s}}_\|\|_3^3, \tag{15}$$

$$\bar{m}_\perp(\bar{\mathbf{s}}_\perp) \;=\; \bar{\mathbf{g}}_\perp^\top \bar{\mathbf{s}}_\perp + \frac{\delta_k}{2} \|\bar{\mathbf{s}}_\perp\|_2^2 + \frac{\mu_k}{3} \|\bar{\mathbf{s}}_\perp\|_3^3. \tag{16}$$

We minimize $\bar{m}_\parallel(\bar{s}_\parallel)$ in (15) similar to how we solved (11). In particular, if we let

$$\mathbf{C}_\parallel = \mathrm{diag}(c_1, \ldots, c_n), \ \text{where } c_i = \frac{2}{\lambda_i + \sqrt{\lambda_i^2 + 4|(\bar{\mathbf{g}}_\parallel)_i|\mu}}, \tag{17}$$

then the solution is given by

$$\mathbf{s}_\parallel^* = -\mathbf{C}_\parallel \bar{\mathbf{g}}_\parallel. \tag{18}$$

Minimizing $\bar{m}_\perp(\bar{s}_\perp)$ in (16) is more challenging. The only restriction on the matrix $\mathbf{U}_\perp$ is that its columns must form an orthonormal basis for the orthogonal complement of the range space of $\mathbf{U}_\parallel$. We are thus free to choose the columns of $\mathbf{U}_\perp$ as long as they satisfy this restriction. In particular, we can choose the first column of $\mathbf{U}_\perp$ to be the normalized orthogonal projection of $\mathbf{g}_k$ onto the orthogonal complement of the range space of $\mathbf{U}_\parallel$, i.e.,

$$(\mathbf{U}_\perp)_1 = (\mathbf{I} - \mathbf{U}_\parallel \mathbf{U}_\parallel^\top)\mathbf{g}_k / \|(\mathbf{I} - \mathbf{U}_\parallel \mathbf{U}_\parallel^\top)\mathbf{g}_k\|_2.$$

If $\mathbf{g}_k \in \mathrm{Range}(\mathbf{U}_\parallel)$, then $\bar{\mathbf{g}}_\perp = \mathbf{U}_\perp^\top \mathbf{g}_k = 0$ and the minimizer of (16) is $\bar{\mathbf{s}}_\perp^* = 0$ (since $\delta_k > 0$ and $\mu_k > 0$). If $\mathbf{g}_k \notin \mathrm{Range}(\mathbf{U}_\parallel)$, then $(\mathbf{U}_\perp)_1 \neq 0$ and we can choose vectors $(\mathbf{U}_\perp)_i \in \mathrm{Range}(\mathbf{U}_\parallel)^\perp$ such that $(\mathbf{U}_\perp)_i^\top(\mathbf{U}_\perp)_1 = 0$ for all $2 \leq i \leq n - k$. Consequently, $\mathbf{U}_\perp^\top(\mathbf{U}_\perp)_1 = \kappa \mathbf{e}_1$, where $\kappa$ is some constant and $\mathbf{e}_1$ is the first column of the identity matrix. Specifically,

$$\kappa \mathbf{e}_1 = \mathbf{U}_\perp^\top(\mathbf{U}_\perp)_1 = \mathbf{U}_\perp^\top \left(\mathbf{U}_\perp \mathbf{U}_\perp^\top \mathbf{g}_k\right) = \mathbf{U}_\perp^\top \mathbf{g}_k = \bar{\mathbf{g}}_\perp,$$

which implies $\kappa = \|\bar{\mathbf{g}}_\perp\|_2$. Thus $\bar{\mathbf{g}}_\perp$ has only one non-zero component (the first component) and therefore, the minimizer $\bar{\mathbf{s}}_\perp^*$ of $\bar{m}_\perp(\bar{\mathbf{s}}_\perp)$ in (16) also has only one non-zero compoent (the first component as well). In particular,

$$(\bar{\mathbf{s}}_\perp^*)_i = \begin{cases} -\alpha^* \|\bar{\mathbf{g}}_\perp\|_2 & \text{if } i = 1 \\ 0 & \text{otherwise} \end{cases},$$

where

$$\alpha = \frac{2}{\delta_k + \sqrt{\delta_k^2 + 4\mu\|\bar{\mathbf{g}}_\perp\|_2}}. \tag{19}$$

Equivalently, $\bar{\mathbf{s}}_\perp^* = -\alpha^* \bar{\mathbf{g}}_\perp$. Note that the quantity $\|\bar{\mathbf{g}}_\perp\|_2$ can be computed without computing $\bar{\mathbf{g}}_\perp$ from the fact that $\|\mathbf{g}\|_2^2 = \|\bar{\mathbf{g}}_\parallel\|_2^2 + \|\bar{\mathbf{g}}_\perp\|_2^2$.

Combining the expressions for $\bar{s}_\parallel^*$ in (18) and for $\bar{\mathbf{s}}_\perp^*$, the solution in the original space is given by

$$\begin{aligned} \mathbf{s}^* &= \mathbf{U}_\parallel \mathbf{s}_\parallel^* + \mathbf{U}_\perp \mathbf{s}_\perp^* \\ &= -\mathbf{U}_\parallel \mathbf{C}_\parallel \mathbf{U}_\parallel^\top \mathbf{g} - \alpha^* (\mathbf{I}_n - \mathbf{U}_\parallel \mathbf{U}_\parallel^\top)\mathbf{g} \\ &= -\alpha^* \mathbf{g} + \mathbf{U}_\parallel(\alpha^* \mathbf{I} - \mathbf{C}_\parallel)\mathbf{U}_\parallel^\top \mathbf{g}. \end{aligned}$$

Note that computing $\mathbf{s}^*$ neither involves forming $\mathbf{U}_\perp$ nor computing $\bar{\mathbf{g}}_\perp$ explicitly.

**Termination criteria.** With each cubic subproblem solved, the iterations are terminated when the change in iterates, $\mathbf{s}_k$, is sufficiently small, i.e.,

$$\|\mathbf{s}_k\|_2 < \tilde{\epsilon}\|\mathbf{y}_k - \mathbf{B}_k\mathbf{s}_k\|_2, \tag{20}$$

for some $\tilde{\epsilon}$, or when the maximum number of allowable iterations is achieved. The proposed Adaptive Regularization using Cubics with L-SR1 (ARCs-LSR1) algorithm is given in Algorithm 1.

**Convergence.** Here, we prove convergence properties of the proposed method (ARCs-LSR1 in Algorithm 1). The following theoretical guarantees follow the ideas from Benson & Shanno (2018); Cartis et al. (2011). First, we make the following mild assumptions:

---

**Algorithm 1** Adaptive Regularization using Cubics with Limited-Memory SR1 (ARCs-LSR1)

---
1: **Given:** $\Theta_0, \gamma_2 \geq \gamma_1, 1 > \eta_2 \geq \eta_1 > 0, \ \sigma_0 > 0, \tilde{\epsilon} > 0, k = 0$, and $k_{\max} > 0$
2: **while** $k < k_{\max}$ and $\|\mathbf{s}_k\|_2 \geq \tilde{\epsilon}\|\mathbf{y}_k - \mathbf{B}_k\mathbf{s}_k\|_2$ **do**
3:      Obtain $\mathbf{S}_k = [\ \mathbf{s}_0 \ \cdots \ \mathbf{s}_k\ ]$ and $\mathbf{Y}_k = [\ \mathbf{y}_0 \ \cdots \ \mathbf{y}_k\ ]$
4:      Solve the generalized eigenvalue problem $\mathbf{S}_k^\top \mathbf{Y}_k \mathbf{u} = \hat{\lambda}\mathbf{S}_k^\top \mathbf{S}_k \mathbf{u}$ and let $\delta_k = \min\{\hat{\lambda}_i\}$
5:      Compute $\boldsymbol{\Psi}_k = \mathbf{Y}_k - \delta_k\mathbf{S}_k$
6:      Perform QR decomposition of $\boldsymbol{\Psi}_k = \mathbf{QR}$
7:      Compute eigendecomposition $\mathbf{RMR}^\top = \mathbf{P\Lambda P}^\top$
8:      Assign $\mathbf{U}_\| = \mathbf{QP}$ and $\mathbf{U}_\|^\top = \mathbf{P}^\top\mathbf{Q}^\top$
9:      Define $\mathbf{C}_\| = \mathrm{diag}(c_1, \ldots, c_k)$, where $c_i = \frac{2}{\lambda_i + \sqrt{\lambda_i^2 + 4\mu|(\bar{\mathbf{g}}_\|)_i|}}$ and $\bar{\mathbf{g}}_\| = \mathbf{U}_\|^\top\mathbf{g}$
10:     Compute $\alpha^*$ in equation 19
11:     Compute step $\mathbf{s}^* = -\alpha^*\mathbf{g} + \mathbf{U}_\|(\alpha^*\mathbf{I} - \mathbf{C}_\|)\mathbf{U}_\|^\top\mathbf{g}$
12:     Compute $m_k(\mathbf{s}^*)$ and $\rho_k = (f(\Theta_k) - f(\Theta_{k+1}))/m_k(\mathbf{s}^*)$
13:     Set

$$\Theta_{k+1} = \begin{cases} \Theta_k + \mathbf{s}^* & \text{if } \rho_k \geq \eta_1 \\ \Theta_k, & \text{otherwise} \end{cases}, \quad \text{and}$$

$$\mu_{k+1} = \begin{cases} \frac{1}{2}\mu_k & \text{if } \rho_k > \eta_2, \\ \frac{1}{2}\mu_k(1 + \gamma_1) & \text{if } \eta_1 \leq \rho_k \leq \eta_2, \\ \frac{1}{2}\mu_k(\gamma_1 + \gamma_2) & \text{otherwise} \end{cases}$$

14:     $k \leftarrow k + 1$
15: **end while**

---

**A1.** The loss function $f(\Theta)$ is continuously differentiable, i.e., $f \in C^1(\mathbb{R}^n)$.

**A2.** The loss function $f(\Theta)$ is bounded below.

Next, we prove that the matrix $\mathbf{B}_k$ in (5) is bounded.

**Lemma 2.1** *The SR1 matrix $\mathbf{B}_{k+1}$ in (5) satsifies*

$$\|\mathbf{B}_{k+1}\|_F \leq \kappa_B \quad \text{for all } k \geq 1$$

*for some $\kappa_B > 0$.*

*Proof:* Using the limited-memory SR1 update with memory parameter $m$ in (5), we have

$$\|\mathbf{B}_{k+1}\|_F \leq \|\mathbf{B}_0\|_F + \sum_{j=k-m+1}^{k} \frac{\|(\mathbf{y}_j - \mathbf{B}_j\mathbf{s}_j)(\mathbf{y}_j - \mathbf{B}_j\mathbf{s}_j)^\top\|_F}{|\mathbf{s}_j^\top(\mathbf{y}_j - \mathbf{B}_j\mathbf{s}_j)|}.$$

Because $\mathbf{B}_0 = \delta_k\mathbf{I}$ with $\delta_k < \delta_{\max}$ for some $\delta_{\max} > 0$, we have that $\|\mathbf{B}_0\|_F = \sqrt{n}\delta_{\max}$. Using a property of the Frobenius norm, namely, for real matrices $\mathbf{A}$, $\|\mathbf{A}\|_F^2 = \mathrm{trace}(\mathbf{AA}^\top)$, we have that $\|(\mathbf{y}_j - \mathbf{B}_j\mathbf{s}_j)(\mathbf{y}_j - \mathbf{B}_j\mathbf{s}_j)^\top\|_F = \|\mathbf{y}_j - \mathbf{B}_j\mathbf{s}_j\|_2^2$. Since the pair $(\mathbf{s}_j, \mathbf{y}_j)$ is accepted only when $|\mathbf{s}_j^\top(\mathbf{y}_j - \mathbf{B}_j\mathbf{s}_j)| > \varepsilon\|\mathbf{s}_j\|_2\|\mathbf{y}_j - \mathbf{B}_j\mathbf{s}_j\|_2$, for some constant $\varepsilon > 0$, and since $\|\mathbf{s}_k\|_2 \geq \tilde{\epsilon}\|\mathbf{y}_k - \mathbf{B}_k\mathbf{s}_k\|_2$, we have

$$\|\mathbf{B}_{k+1}\|_F \leq \sqrt{n}\delta_{\max} + \frac{m}{\varepsilon\tilde{\epsilon}} \equiv \kappa_B,$$

which completes the proof. $\square$

Given the bound on $\|\mathbf{B}_{k+1}\|_F$, we obtain the following result, which is similar to Theorem 2.5 in Cartis et al. (2011).

**Theorem 2.2** *Under Assumptions **A1** and **A2**, if Lemma 2.1 holds, then*

$$\liminf_{k \to \infty} \|\mathbf{g}_k\| = 0.$$

Finally, we consider the following assumption, which can be satisfied when the gradient, $\mathbf{g}(\Theta)$, is Lipschitz continuous on $\Theta$.

**A3.** If $\{\Theta_{t_i}\}$ and $\{\Theta_{l_i}\}$ are subsequences of $\{\Theta_k\}$, then $\|\mathbf{g}_{t_i} - \mathbf{g}_{l_i}\| \to 0$ whenever $\|\Theta_{t_i} - \Theta_{l_i}\| \to 0$ as $i \to \infty$.

If we further make Assumption **A3**, we have the following stronger result (which is based on Corollary 2.6 in Cartis et al. (2011)):

**Corollary 2.3** *Under Assumptions **A1**, **A2**, and **A3**, if Lemma 2.1 holds, then*

$$\lim_{k \to \infty} \|\mathbf{g}_k\| = 0.$$

By Corollary 2.3, the proposed ARCs-LSR1 method converges to first-order critical points.

**Stochastic implementation.** Because full gradient computation is very expensive to perform, we impement a stochastic version of the proposed ARCs-LSR1 method. In particular, we use the batch gradient approximation

$$\tilde{\mathbf{g}}_k \equiv \frac{1}{|\mathcal{B}_k|} \sum_{i \in \mathcal{B}_k} \nabla f_i(\Theta_k).$$

In defining the SR1 matrix, we use the quasi-Newton pairs $(\mathbf{s}_k, \tilde{\mathbf{y}}_k)$, where $\tilde{\mathbf{y}}_k = \tilde{\mathbf{g}}_{k+1} - \tilde{\mathbf{g}}_k$ (see e.g., Erway et al. (2020)). We make the following additional assumption (similar to Assumption 4 in Erway et al. (2020)) to guarantee that the loss function $f(\Theta)$ decreases over time:

**A4.** The loss function $f(\Theta)$ is fully evaluated at every $J > 1$ iterations (for example, at iterates $\Theta_{J_0}, \Theta_{J_1}, \Theta_{J_2}, \ldots$, where $0 \le J_0 < J$ and $J = J_1 - J_0 = J_2 - J_1 = \cdots$) and nowhere else in the algorithm. The batch size $d$ is increased monotonically if $f(\Theta_{J_\ell}) > f(\Theta_{J_{\ell-1}}) - \tau$ for some $\tau > 0$.

With this added assumption, we can show that the stochastic version of the proposed ARCs-LSR1 method converges.

**Theorem 2.4** *The stochastic version of ARCs-LSR1 converges with*

$$\lim_{k \to \infty} \|\mathbf{g}_k\| = 0.$$

*Proof:* Let $\widehat{\Theta}_i = \Theta_{J_i}$. By Assumption 4, $f(\Theta)$ must decrease monotonically over the subsequence $\{\widehat{\Theta}_i\}$ or $d \to |\mathcal{D}|$, where $|\mathcal{D}|$ is the size of the dataset. If the objective function is decreased $\iota_k$ times over the subsequence $\{\widehat{\Theta}_i\}_{i=0}^k$, then

$$f(\widehat{\Theta}_k) = f(\hat{\Theta}_0) + \sum_{i=1}^{\iota_k} \left\{ f(\widehat{\Theta}_i) - f(\widehat{\Theta}_{i-1}) \right\} \le f(\widehat{\Theta}_0) - \iota_k \tau.$$

If $d \to |\mathcal{D}|$, then $\iota_k \to \infty$ as $k \to \infty$. By Assumption **A2**, $f(\Theta)$ is bounded below, which implies $\iota_k$ is finite. Thus, $d \to |\mathcal{D}|$, and the algorithm reduces to the full ARCs-LSR1 method, whose convergence is guaranteed by Corollary 2.3. $\square$

We note that the proof to Theorem 2.4 follows very closely the proof of Theorem 2.2 in Erway et al. (2020).

**Complexity analysis.** SGD methods and the related adaptive methods require $\mathcal{O}(n)$ memory storage to store the gradient and $\mathcal{O}(n)$ computational complexity to update each iterate. Such low memory and

computational requirements make these methods easily implementable. Quasi-Newton methods store the previous $m$ gradients and use them to compute the update at each iteration. Consequently, L-BFGS methods require $\mathcal{O}(mn)$ memory storage to store the gradients and $\mathcal{O}(mn)$ computational complexity to update each iterate (see Burdakov et al. (2017) for details). Our proposed ARCs-LSR1 approach also uses $\mathcal{O}(mn)$ memory storage to store the gradients, but the computational complexity to update each iterate requires an additional eigendecomposition of the $m \times m$ matrix $\mathbf{RMR}^\top$, so that the overall computational complexity at each iteration is $\mathcal{O}(m^3 + mn)$. However, since $m \ll n$, this additional factorization does not significantly increase the computational time.

## 3 Results

**Optimization approaches.** We list the various optimization approaches to which we compared our proposed method. For the numerical experiments, we empirically fine-tuned the hyperparameters and selected the best for each update scheme.

1. **Stochastic Gradient Descent (SGD) with Momentum** (see e.g., Qian (1999)). For the experiments, we used a momentum parameter of 0.9 and a learning rate of $1.0 \times 10^{-1}$.

2. **Adaptive Gradient Algorithm (Adagrad)** (see Duchi et al. (2011)). In our experiments, the initial accumulator value is set to 0, the perturbation $\epsilon$ is set to $1.0 \times 10^{-10}$, and the learning rate is set to $1.0 \times 10^{-2}$.

3. **Root Mean Square Propagation (RMSProp)** (see Hinton et al. (2012)). For our experiments, the perturbation $\epsilon$ is set to $1.0 \times 10^{-8}$. We set $\alpha = 0.99$, and used a learning rate of $1.0 \times 10^{-2}$.

4. **Adam** (see Kingma & Ba (2014)). For our experiments, we apply an $\epsilon$ perturbation of $1.0 \times 10^{-6}$. The momentum parameters $\beta_0$ and $\beta_1$ are chosen to be 0.9 and 0.999, respectively. The learning rate is set to $1.0 \times 10^{-3}$.

5. **Limited-memory BFGS (L-BFGS)**: We set the default learning rate to 1.0. The tolerance on function value/parameter change is set to $1.0 \times 10^{-9}$ and the first-order optimality condition for termination is defined as $1.0 \times 10^{-9}$

6. **ARCs-LSR1 (Proposed method)**: For the experiments, we choose the same parameters as those used in L-BFGS.

**Dataset.** We measure the performance of each optimization method on the following five commonly-used datasets for training and testing in machine learning: (1) MNIST (LeCun et al., 2010), (2) CIFAR10 (Krizhevsky et al., 2010), (3) Fashion-MNIST (Xiao et al., 2017) and (4) IRIS (Fisher, 1936; Anderson, 1935) and (5) Penn Tree Bank (Marcus et al., 1993).

To empirically compare the efficiency of the proposed method with widely-used optimization methods, we focus on three broad deep learning problems: image classification, image reconstruction and language modeling. All experiments were conducted using open-source software PyTorch (Paszke et al., 2019), SciPy (Virtanen et al., 2020), and NumPy (Harris et al., 2020). We use an Intel Core i7-8700 CPU with a clock rate of 3.20 GHz and an NVIDIA RTX 2080 Ti graphics card.

### 3.1 Experiment I: Image classification

We present the classification results for IRIS, MNIST, and CIFAR.

**Experiment I.A: IRIS.** This dataset is relatively small; consequently, we only consider a shallow network with three fully connected layers and 2953 parameters. We set the history size and maximum iterations for the proposed approach and L-BFGS to 10. Figure 1(a) shows the comparative performance of all the methods. Note that our proposed method (ARCs-LSR1) achieves the highest classification accuracy in the fewest number of epochs.

**Experiment I.B: MNIST.** The MNIST classifier is a shallow network with 3 fully connected layers and 397510 parameters. We train the network for 20 epochs with a batch size of 256 images, keeping the history size and maximum iterations same for the proposed approach and L-BFGS. Figure 2(a) shows that the proposed ARCs-LSR1 outperforms the other methods.

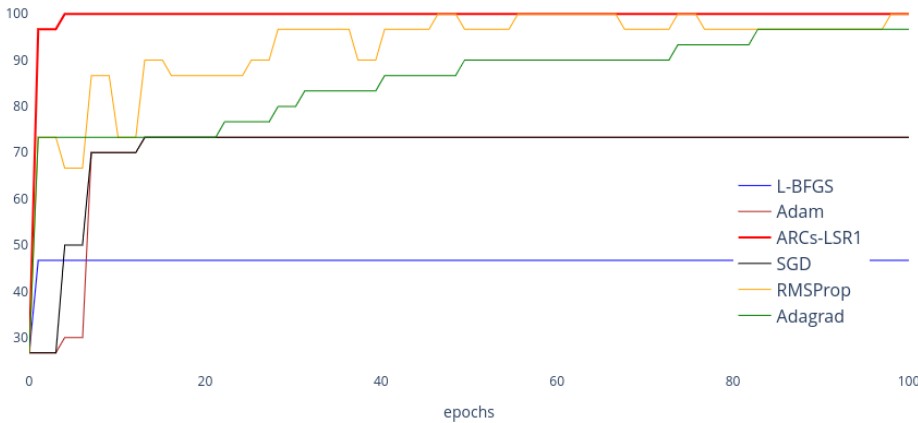

Figure 1: The classification accuracy results for **Experiment I.A: IRIS**. The percentage of testing samples correctly predicted in the testing dataset for each method is presented. Note that the proposed method (ARCs-LSR1) achieves the highest classification accuracy within the fewest number of epochs.

**Experiment I.C: CIFAR10.** Because the CIFAR10 dataset contains color images (unlike the MNIST grayscale images), the network used has more layers compared to the previous experiments. The network has 6 convolutional layers and 3 fully connected layers with 62006 parameters. For ARCs-LSR1 and L-BFGS, we have a history size of 100 with a maximum number of iterations of 100 and a batch size of 1024. Figure 2(b) represents the testing accuracy, i.e., the number of samples correctly predicted in the testing set.

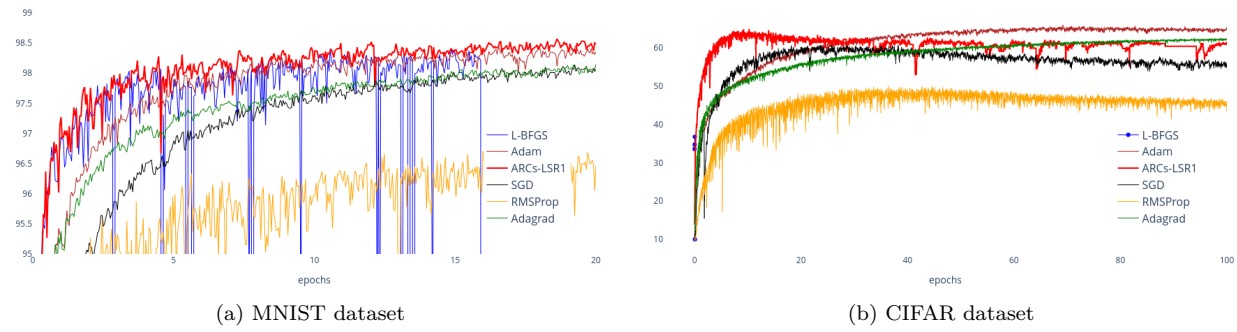

|  (a) MNIST dataset | (b) CIFAR dataset |
| --- | --- |

Figure 2: The classification accuracy results for **Experiment I.B and I.C**. The percentage of testing samples correctly predicted in the testing dataset for each method is presented. Note that the proposed method (ARCs-LSR1) achieves the highest classification accuracy within the fewest number of epochs.

**Experiment I: Additional MNIST results.** We select the best history, max-iterations and batch-size hyperparameters by conducting a thorough parameter search described below.

**History.** The ARCs-LSR1 method requires some history from the past to form the Limited-memory SR1 approximation. The history stores a set of steps '**s**' and their corresponding change in gradients '**y**'. This is the most important parameter for the proposed approach - as the number of history pairs increase, the approximation begins converging to the true Hessian. However, we cannot have a full-rank approximation, so the number of history pairs are limited. In addition, in the context of deep learning, a high memory parameter might not be ideal owing to its large storage complexity.

To empirically show this, we ran a set of experiments by varying the batch-size and the the number of iterations for each batch and present results on the MNIST classification task. We selected a history-size of $5, 10, 15, 20, 50$ and $100$.

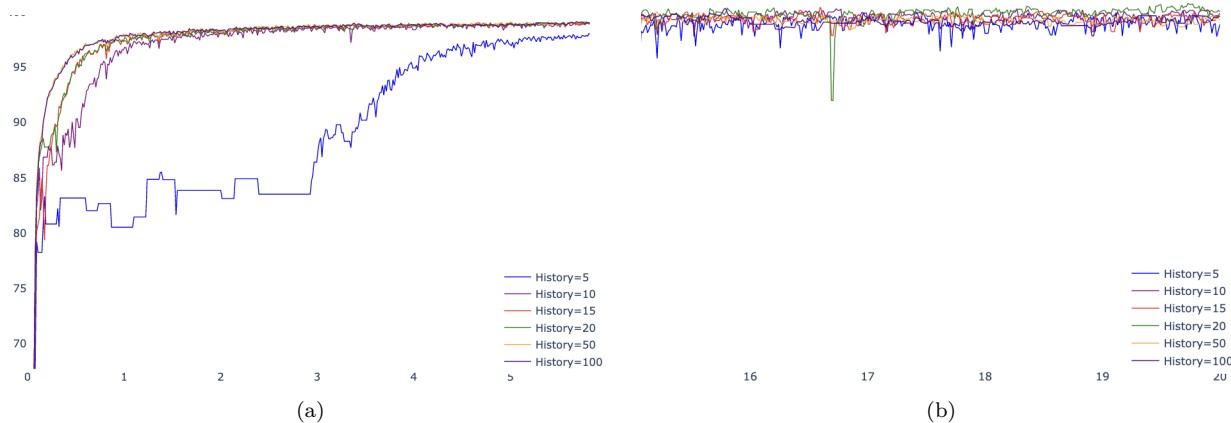

(a)          (b)

Figure 3: **MNIST classification.** We fix the maximum iterations to 1 and batch-size of 128. (a) presents the epochs [1-5] and (b) presents epochs [15-20].

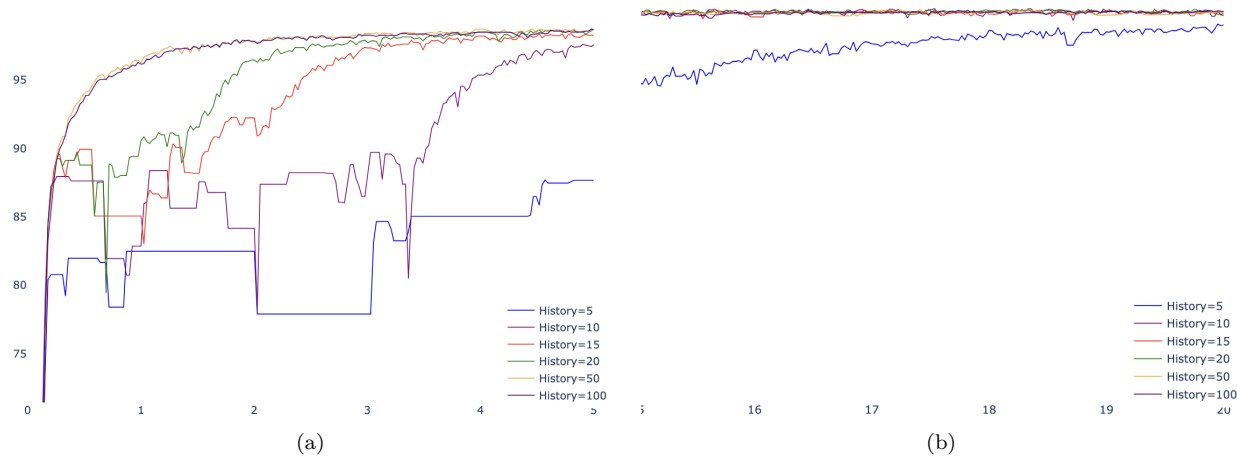

(a)          (b)

Figure 4: **MNIST classification.** We fix the maximum iterations to 1 and batch-size of 256. (a) presents the epochs [1-5] and (b) presents epochs [15-20].

**Different max-iterations.** The max-iterations determines how many times the proposed approach is applied to each individual batch for an optimization step and its corresponding history update $(\mathbf{s}, \mathbf{y})$. For the most ideal condition, we consider the trade-off between computational complexity and improvement of accuracy. This means the accuracy of prediction does not increase significantly with the increase in the upper bound of iterations. We fix the batch-size to 128 and switch the max-iterations between $10, 15$ and $20$.

The results are presented in Figure(s) 7, 8, and 9. From these results, it was certain that a maximum iteration of 10 was ideal.

**Different batch-sizes.** For this experiment, we chose from batch-sizes of $128, 256, 512$ and $1024$. We fixed the maximum-iterations to 1. The results are presented in Figure(s) 3,4, 5, and 6.

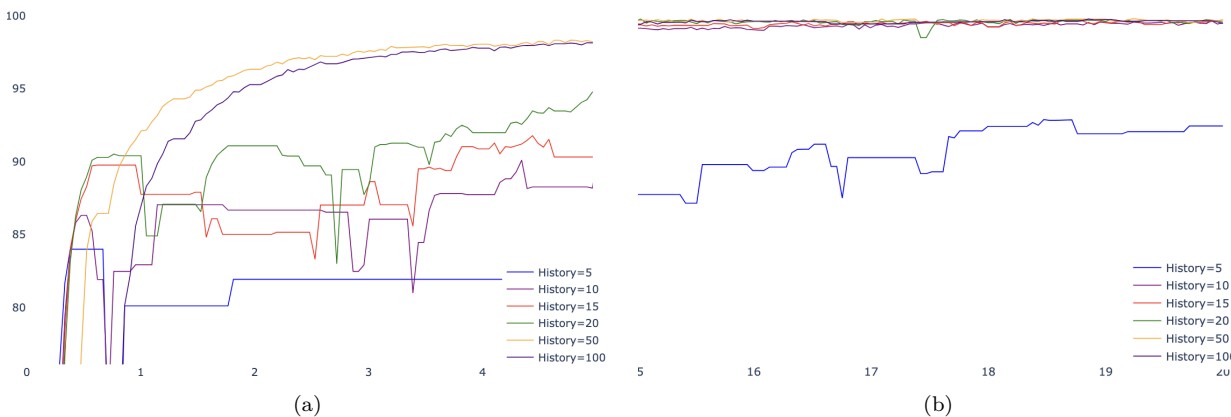

Figure 5: **MNIST classification.** We fix the maximum iterations to 1 and batch-size of 512. (a) presents the epochs [1-5] and (b) presents epochs [15-20].

## 3.2 Experiment II: Image reconstruction

The image reconstruction problem involves feeding a feedforward convolutional autoencoder model a batch of the dataset. The loss function is defined between the reconstructed image and the original image. We use the Mean-Squared Error (MSE) loss between the reconstructed image and the original image. For this experiment, we use the MNIST and FMNIST dataset.

**Experiment II.A: MNIST.** The network is shallow, with 53415 parameters, which are initialized randomly. We considered a batch size of 256 images and trained over 50 epochs. Each experiment has been conducted 5 times. The results for the image reconstruction can be seen in Figure 11, where the initial descent of the proposed approach yields a significant decrease in the training loss. We provide the training loss results for the early (Figure 11(a)) and late epochs (Figure 11(b)). In Figure 11(b), all the methods eventually converge to the same training loss value (except for L-BFGS). We see a similar trend during the early and late epochs for the testing loss (see Figure 12).

**Experiment II: FMNIST.** Figure(s) 10(a) and 10(b) show the testing response for the early and late epochs, respectively. The early iterates generated by the proposed approach significantly decreases the

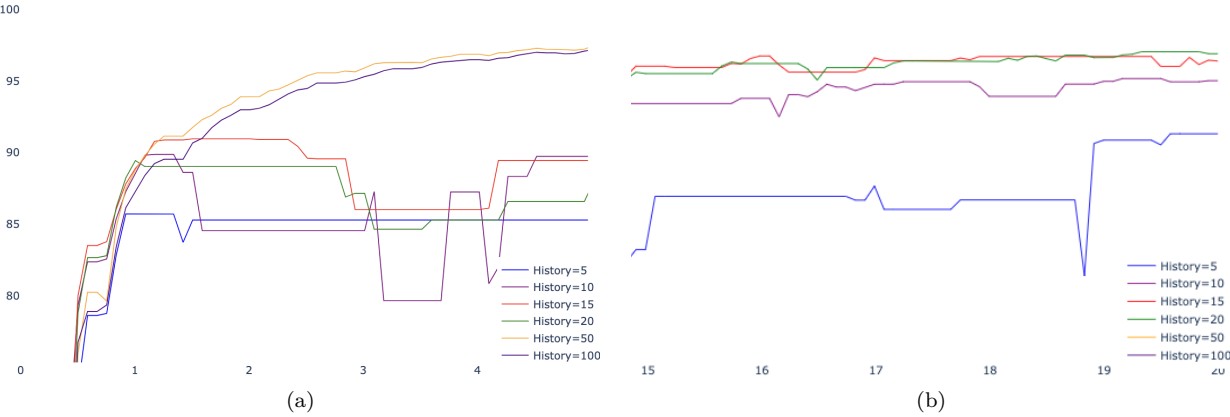

Figure 6: **MNIST classification.** We fix the maximum iterations to 1 and batch-size of 1024. (a) presents the epochs [1-5] and (b) presents epochs [15-20].

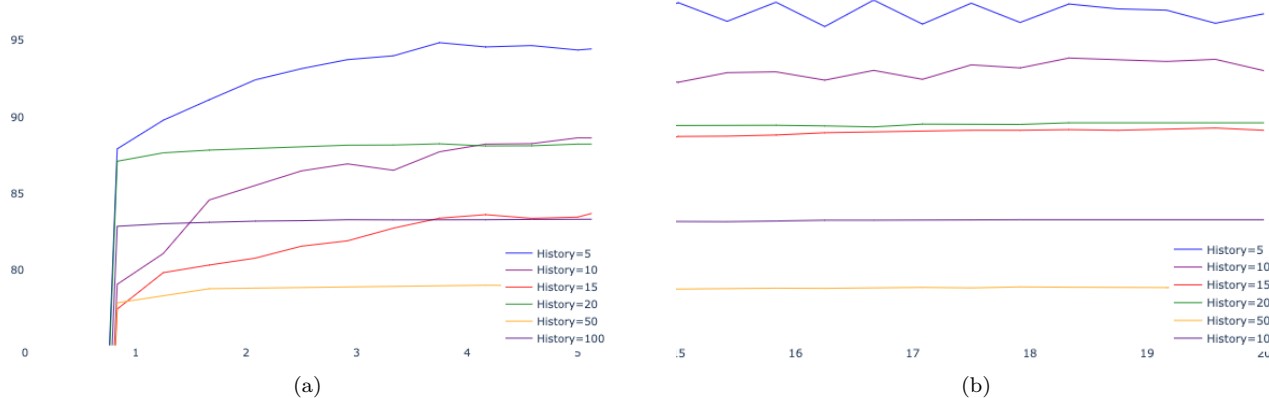

Figure 7: **MNIST classification:** The figure shows the classification response for a upper bound max-iterations of 10. The batch-size is fixed to 128 images. (a) presents the early epochs $[1-5]$ while the second column (b) presents the late epochs $[15-20]$.

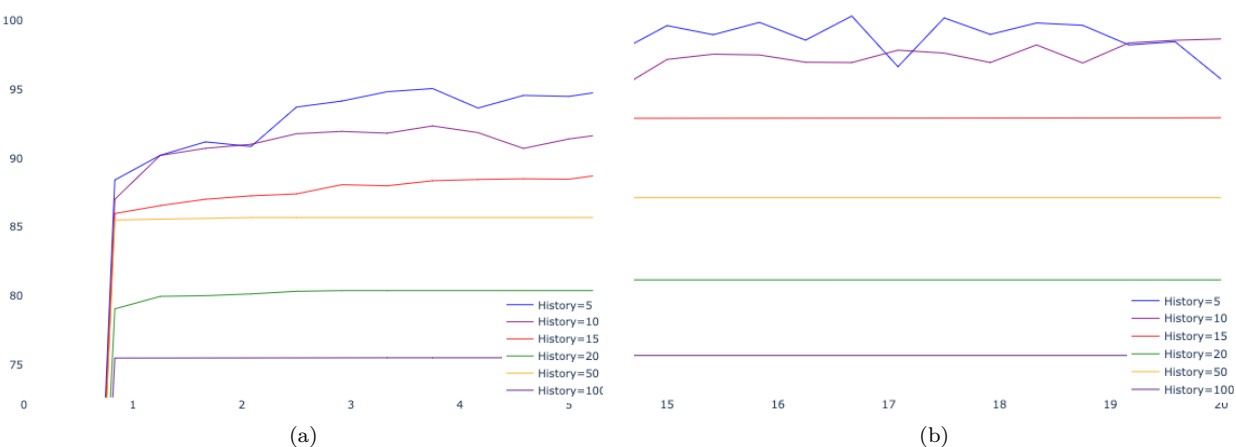

Figure 8: **MNIST classification:** The figure shows the classification response for a upper bound max-iterations of 15. The batch-size is fixed to 128 images. (a) presents the early epochs $[1-5]$ while the second column (b) presents the late epochs $[15-20]$.

objective function. The proposed approach has maintained this trend in the later epochs as well (see Figure 10(b)). This shows that the network is capable of generalizing on a testing dataset as well in comparison to all other adaptive and quasi-Newton methods.

## 3.3 Experiment III: Natural language modeling

We conducted word-level predictions on the Penn Tree Bank (PTB) dataset (Marcus et al., 1993). We used a state-of-the-art Long-Short Term Memory (LSTM) network which has 650 units per layer and its parameters are uniformly regularized in the range [-0.05, 0.05]. For more details on implementation, please refer Zaremba et al. (2014). For the ARCs-LSR1 method and the L-BFGS, we used a history size of 5 over 4 iterations. The prediction loss results are shown in Figure 13. In contrast to the previous experiments, here, both quasi-Newton methods (L-BFGS and ARCs-LSR1) outperform the adaptive methods, with the proposed method (ARCs-LSR1) achieving the lowest cross-entropy prediction loss.

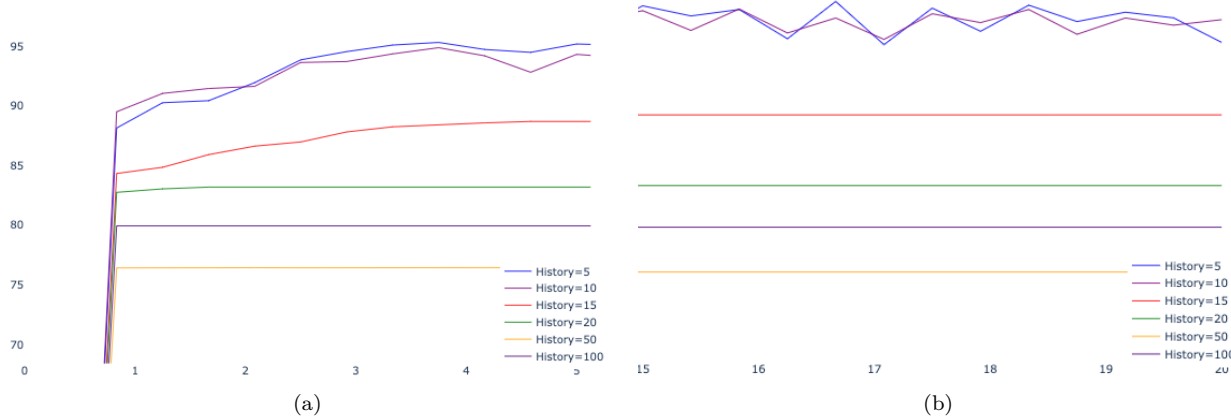

(a)                                          (b)

Figure 9: **MNIST classification:** The figure shows the classification response for a upper bound max-iterations of 20. The batch-size is fixed to 128 images. (a) presents the early epochs $[1 - 5]$ while the second column (b) presents the late epochs $[15 - 20]$.

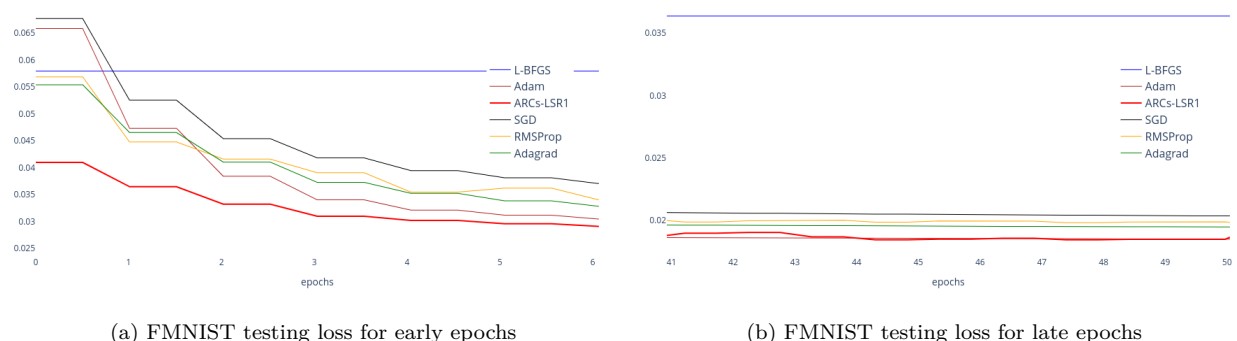

(a) FMNIST testing loss for early epochs            (b) FMNIST testing loss for late epochs

Figure 10: The image reconstruction results for **Experiment II**. (a) Initial testing loss of the network. The $y$-axis represents the MSE loss in the first 6 epochs. (b) The final MSE loss of the testing samples from epoch 41 to 50. The proposed method (ARCs-LSR1) achieves the lowest testing loss.

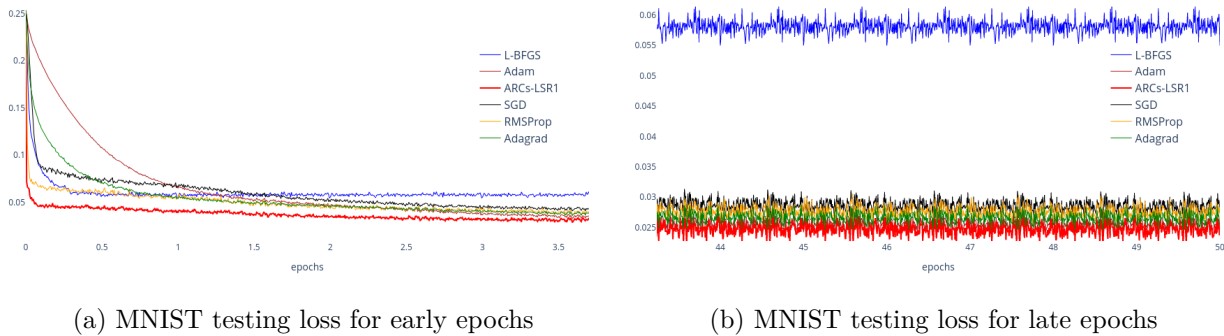

(a) MNIST testing loss for early epochs             (b) MNIST testing loss for late epochs

Figure 11: The image reconstruction results for **Experiment II.A: MNIST**. (a) Initial training loss. The $y$-axis represents the Mean-Squared Error (MSE) loss from the first four epochs. (b) Final training loss from epochs 43 to 50. Note that the proposed method (ARCs-LSR1) achieves the lowest training loss.

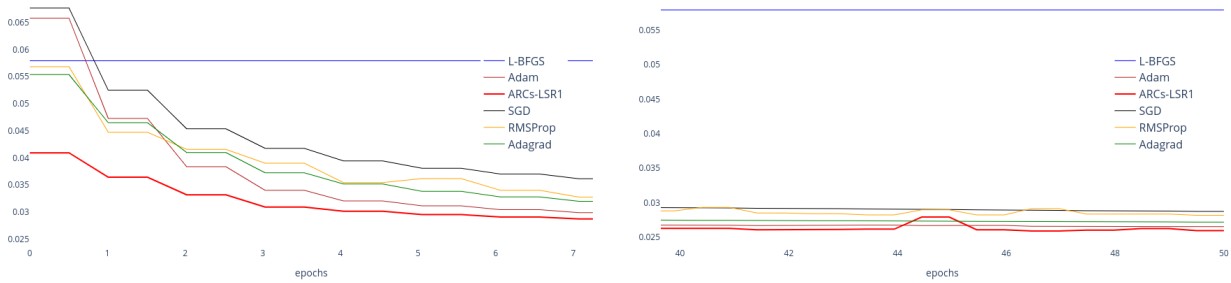

(a) MNIST training loss for early epochs         (b) MNIST training loss for late epochs

Figure 12: The image reconstruction results for **Experiment II.A: MNIST**. (a) Initial testing loss. The $y$-axis represents the Mean-Squared Error (MSE) loss from the first four epochs. (b) Final testing loss from epochs 43 to 50. Note that the proposed method (ARCs-LSR1) achieves the lowest testing loss.

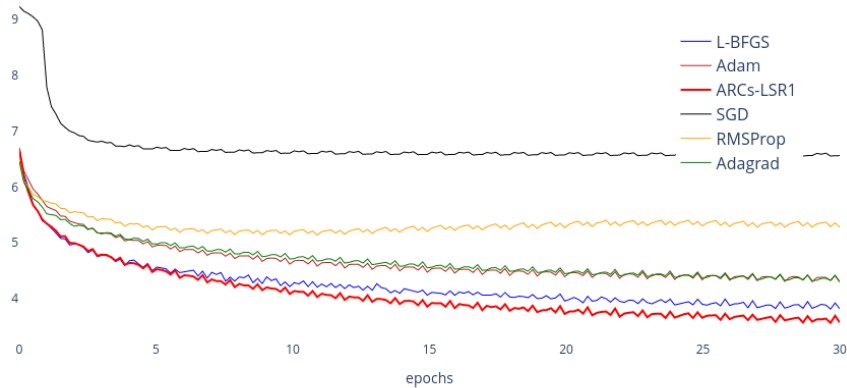

Figure 13: The prediction loss for **Experiment III: Penn Tree Bank**. The $y$-axis represents the cross-entropy loss, and the $x$-axis represents the number of epochs. Note that the proposed method (ARCs-LSR1) achieves the lowest loss.

### 3.4 Experiment IV: Comparison with Stochastically Damped L-BFGS

In the previous experiments on image classification and reconstruction (Experiments I and II), the L-BFGS approach performs poorly, which can be attributed to noisy gradient estimates and non-convexity of the problems. To tackle this, a *stochastically damped* L-BFGS (SdLBFGS) approach was proposed (see Wang et al. (2017)) which adaptively generates a variance reduced, positive-definite approximation of the Hessian. We compare the proposed approach to L-BFGS and SdLBFGS on the MNIST classification problem. From Figure 14(a), the proposed approach achieves a comparable performance to the stochastic version and is able to achieve the best accuracy in later epochs (see Figure 14(b)).

### 3.5 Experiment V: Timing results

We take the CIFAR10 experiment into consideration as its the most computationally expensive experiment for classification with a parameter count of 62k and a memory parameter of 100, with a total of $\approx$ 6M memory allocations. Figure 15 shows the time-budget for each of the adaptive techniques and the proposed approach. We observe that the proposed approach is able to achieve the highest accuracy the quickest, even with a higher computational budget.

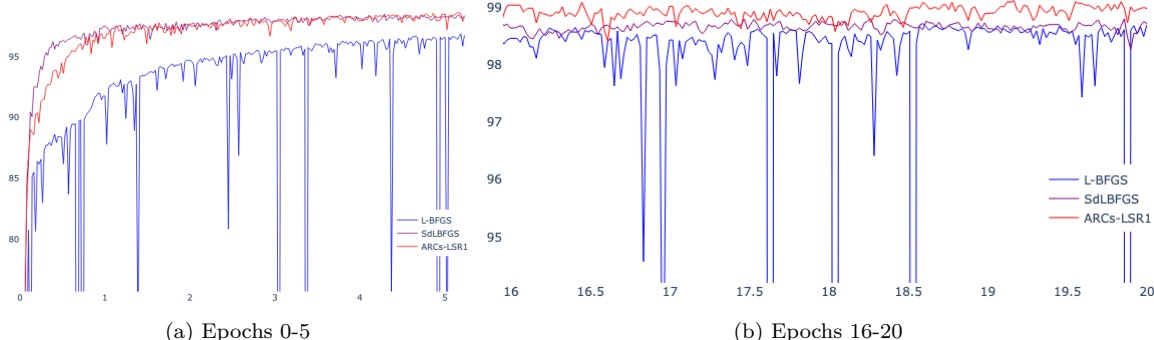

(a) Epochs 0-5            (b) Epochs 16-20

Figure 14: The prediction loss for **Experiment IV**: Comparison with stochastically damped L-BFGS. The $x$-axis represents the number of epochs and the $y$-axis represents the accuracy of prediction. (a) Accuracy for epochs 0-5. (b) Accuracy for epochs 16-20.

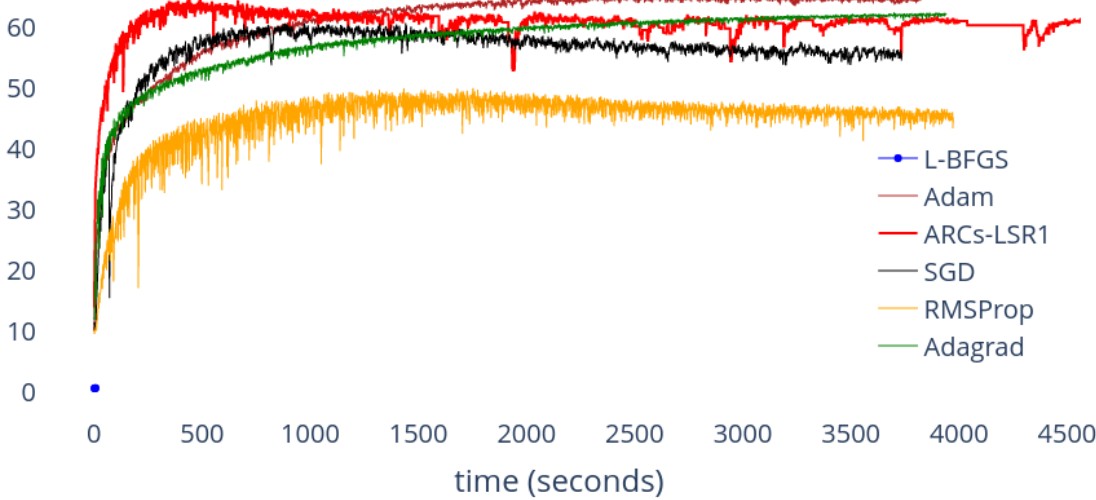

Figure 15: **Experiment V**: CIFAR-10 classification time complexity. The figure shows the time complexity of all the methods and the proposed approach. Even though the proposed approach takes $\approx 500$ seconds longer, the best accuracy is achieved the fastest in comparison to all the other state-of-the-art approaches.

## 4 Conclusion

In this paper, we proposed a novel quasi-Newton approach in an adaptive regularized cubics (ARCs) setting using the less frequently used limited-memory Symmetric Rank-1 (L-SR1) update and a shape-changing norm to define the regularizer. This shape-changing norm allowed us to solve for the minimizer exactly. We provided convergence guarantees for the proposed ARCs-LSR1 method and analyzed its computational complexity. Using a set of experiments in classification, image reconstruction, and language modeling, we demonstrated that ARCs-LSR1 achieves the highest accuracy in fewer epochs than a variety of existing state-of-the-art optimization methods. Striking a comfortable balance between the computational and space

complexity, the competitive nature of the ARCs-LSR1 performance makes it a superior alternative to existing gradient and quasi-Newton based approaches.

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
