# OpenReview forum: "Symmetric Rank-One Quasi-Newton Methods for Deep Learning Using Cubic Regularization"
_TMLR — Withdrawn by Authors_

### Review · Reviewer_AsiY · 2025-03-16

**Summary Of Contributions:**

This paper proposes a quasi-Newton method that allows indefinite Hessian approximations, enabling directions of negative curvature to be exploited. A cubic regularization is used to ensure that the resulting search direction is a descent direction.

**Audience:**

Yes

**Broader Impact Concerns:**

This is a study on optimizers and has no direct broader implications.

**Claims And Evidence:**

Yes

**Requested Changes:**

Please add experiments with ResNet18 on CIFAR100 and nanoGPT on WikiText to show that the proposed optimizer shows an advantage for more modern model architectures. These are small scale experiments so it should not be too much to ask for.

Please show how the optimizers perform when the hyperparameters are perturbed.

Figures 2(b) and 15 should have grid lines, since it is difficult to see whether the best accuracy of ARCs-LSR1 is better than Adam.

**Strengths And Weaknesses:**

Strengths
Experimental results how that the proposed optimizer ARCs-LSR1 converges much faster than the baselines such as SGD, Adagrad, RMSProp, Adam, L-BFGS, not just in terms of the number of epochs, but also with respect to the training time. Experiments are performed on various datasets (MNIST, CIFAR10, Fashion-MNIST, IRIS, Penn Tree Bank).

Weaknesses
The models used in the experiments (3 fully connected layers, 6 convolutional and 3 fully connected layers) are very primitive. Improvements in the model architecture such as batch normalization are known to diminish the advantage of second order optimizers because they make the loss landscape better conditioned. The conclusions drawn from such primitive model architectures are unlikely to generalize to more modern model architectures.

Experiments seem to be performed using fixed hyperparameters, and it is unclear whether SGD, Adagrad, RMSProp, Adam, L-BFGS, and ARCs-LSR1 were all using optimal hyperparameters. These optimizers have different sensitivity to hyperparameters, which is an important factor when choosing optimizers. Therefore, it is important to show how these optimizers perform when the hyperparameters are perturbed.

ARCs-LSR1 seems to have a stronger tendency to overfit compared to other optimizers, as can be seen from Fig. 2 (b) and Fig. 15. This may not be an issue when training multiple epochs on small datasets like MNIST and CIFAR, but would be a disadvantage for single epoch training of large language models where early stopping leads to smaller coverage of data.

A perspective that is missing from the paper is that optimization in machine learning is less about convergence and more about generalization. An optimizer that converges faster is not necessarily the best choice. For datasets like MNIST and CIFAR where the training and test data have very similar distributions, it is difficult to measure the generalization aspect of the optimizer.

---

### Review · Reviewer_UiHo · 2025-03-26

**Summary Of Contributions:**

The authors develop a low-memory quasi-Newton method based on combining an SR1-style update with Adaptive Cubic Regularization (ARC) for training neural networks.
As the Hessian of neural networks is generally indefinite due to non-convexity, without using ARC, the step taken by an SR1-update cannot be guaranteed to be a descent direction.
Moreover, by choosing the norm in the ARC penalty appropriately, they are able to reduce a normally challenging subproblem to one that has an inexpensive closed form solution.
The paper establishes under standard assumptions that the proposed approach converges to a stationary point both for the full and batch gradient setting.
Finally the authors conduct several numerical experiments on MLP, autoencoder, and RNN architectures to verify the efficacy of the approach.

**Audience:**

Yes

**Broader Impact Concerns:**

As the paper focuses on developing a new method for Deep Learning optimization, it does not pose any direct societal or ethical implications.
Thus, a broader impact statement is unnecessary.

**Claims And Evidence:**

Yes

**Requested Changes:**

I have three main suggestions for revising the work.
The first two are essential in order for me to recommend acceptance.
The third is a recommendation based on what I think could be a good application of the proposed algorithm.

**(1) Related Work**

A more thorough related work section must be added to the paper. Briefly discussing some of the ideas in / citing the papers I mentioned above would be sufficient in this regard.
In my view, this is essential as it helps the reader place the proposal in the context of what is now a very big literature.

**(2) Experiments**

In the numerical experiments, the paper needs to add a comparison with methods like Shampoo or AdaHessian, preferably both.
The comparison is needed to show how the proposal compares to popular sophisticated optimizers for deep learning.
This should not be too burdensome as both methods have readily available open source implementations.

In addition to adding these comparisons, I would like the authors to have at least two experiments where the learning rate of the
competing optimizers was tuned. You can tune your as well if you like.

Having a comparison against tuned competitors is really essential, as in practice learning rate tuning is essentially default practice.
I should emphasize it is fine if you don't win against the tuned versions, this is quite hard, and is partly why first-order methods remain the
most popular.
It's important that the proposed method be not that much worse.

**(3) A potential application**

The authors might consider replacing one or two of their experimental problems, with problems involving Physics Informed Neural Networks (PINNs).
PINNs are known to be very challenging to optimize.
Indeed, often state-of-the-art training algorithms employ L-BFGS or Newton type methods in the training pipeline.
In addition, the networks in PINNs tend to be shallower and have less parameters than the networks deployed in contemporary deep learning.
Thus, I believe they may provide an ideal application for the method developed in this paper.
Good performance on PINN problem or two that is known to be challenging would greatly strengthen the case for the usefulness of the method.
If the authors choose to go down this route, they can look at the papers [1] and [2] for PINN problems that are known to be challenging to optimize.

I want to emphasize again this is simply a suggestion, if the authors choose not to follow it, it will not impact my final decision regarding the paper.

**References**

[1] Krishnapriyan, Aditi, Amir Gholami, Shandian Zhe, Robert Kirby, and Michael W. Mahoney. "Characterizing possible failure modes in physics-informed neural networks." Advances in Neural Information Processing Systems 34 (2021): 26548-26560.

[2] Rathore, Pratik, Weimu Lei, Zachary Frangella, Lu Lu, and Madeleine Udell. "Challenges in Training PINNs: A Loss Landscape Perspective." In International Conference on Machine Learning, pp. 42159-42191. PMLR, 2024.

**Strengths And Weaknesses:**

**Strengths**
- I have not seen a this combination of SR1 with ARC before, so the approach taken is new. Moreover, thanks to a judicious choice of the penalty, solving the subproblem is inexpensive. This circumnavigates a common issue with cubic regularized methods, where the subproblem is too expensive to solve in a reasonable enough amount of time to remain competitive with first-order methods.
- The authors show the iterates of the proposed method approach a stationary point.
This is essentially the best that can hoped for without making further regularity assumptions.
- The experiments show the approach has some promise.
- The key ideas of the paper are presented clearly and are easy to follow.

**Weaknesses**

The main approach is interesting, and likely to be of interest to certain sectors of the TMLR community.
However, the paper at present has some issues that need to be addressed.

*Related work*
- A major weakness of the paper is a lack of a dedicated related work section.
Much work has been done on deep learning optimization beyond first-order methods, and the paper discusses none of it.
In particular, this paper is not the first to use cubic-regularization/ARC style methods in optimization for machine learning or deep ;earning.
Two seminal papers in this direction are [1] and [2]., both of which must be cited.
In addition, the paper ought to cite some more modern methods based on randomized diagonal and low-rank approximations to the Hessian [3,4], as well as methods based on gradient whitening such as Shampoo [5].
A brief discussion of the differing approaches is essential so that readers can relate this work to the existing literature.

*Experiments*

Overall the experiments aren't very compelling for several reasons:

i) Lack of hyperparameter tuning: None of the competing methods in the experiments were tuned for their best performance.
In deep learning it is standard practice to select Adam or SGD's learning rate through a grid search.

ii) Lack of comparison with modern methods: The paper does not make the best choice of methods for comparison.
Comparing with AdaGrad and RMSProp isn't very interesting comparisons as these methods are no longer close to state-of-the-art and are rarely used in practice.
It would be better to compare to more modern stochastic first-order methods like Yogi or D-Adapted Adam.

In addition, the comparison with contemporary second-order or more advanced first-order preconditioning methods is missing.
The paper should include a comparison with a method like AdaHessian or Shampoo.
A comparison like this would show the method can compete with the current popular alternatives to Adam and SGD, which like the proposed method are also generally more computationally expensive.

iii) Lack of a learning rate schedule
This is a more minor point, but in practice the best performance is typically achieved by using a learning rate schedule like cosine decay. I'm curious why the authors chose not to use a schedule, as it might also make their method perform better

**References**

[1] Kohler, Jonas Moritz, and Aurelien Lucchi. "Sub-sampled cubic regularization for non-convex optimization." In International Conference on Machine Learning, pp. 1895-1904. PMLR, 2017.

[2] Tripuraneni, Nilesh, Mitchell Stern, Chi Jin, Jeffrey Regier, and Michael I. Jordan. "Stochastic cubic regularization for fast nonconvex optimization." Advances in Neural Information Processing Systems 31 (2018).

[3] Yao, Zhewei, Amir Gholami, Sheng Shen, Mustafa Mustafa, Kurt Keutzer, and Michael Mahoney. "Adahessian: An adaptive second order optimizer for machine learning." In Proceedings of the AAAI conference on Artificial Intelligence, vol. 35, no. 12, pp. 10665-10673. 2021.

[4] Frangella, Zachary, Pratik Rathore, Shipu Zhao, and Madeleine Udell. "Sketchysgd: reliable stochastic optimization via randomized curvature estimates." SIAM Journal on Mathematics of Data Science 6, no. 4 (2024): 1173-1204.

[6] Gupta, Vineet, Tomer Koren, and Yoram Singer. "Shampoo: Preconditioned stochastic tensor optimization." In International Conference on Machine Learning, pp. 1842-1850. PMLR, 2018.

---

### Review · Reviewer_9yvA · 2025-06-30

**Summary Of Contributions:**

This paper proposes a modified adaptive regularized cubics approach with a limited-memory symmetric rank-one quasi-Newton approach for the non-convex loss function. The convergence of the proposed algorithm ARCs-LSR1 is proven, and experiments of ARCs-LSR1 are conducted, which validates its performance.

**Audience:**

Yes

**Claims And Evidence:**

Yes

**Requested Changes:**

#Questions
1) Besides the Adaptive Regularization using Cubics (ARCs) method, are there other regularization methods that can be combined with SR1 Hessian approximation? If so, why is the ARCs method preferable? Would additional comparative experiments using different regularization terms better demonstrate the advantages of ARCs-LSR1?
2) In the image reconstruction experiments, only the MSE loss function is used. If you conduct experiments with more loss functions, will they have different impacts on ARCs-LSR1?

#Suggestions
1) In Page 3, the matrix $\mathbf{M}$ in $\mathbf{RMR}^\top$ is not explicitly defined. However, since its definition is provided in Erway et al. (2020), this issue can be easily resolved.
2) Only the boldface notation $\mathbf{s}$ is used in Equations (15), (16), and (18), but subsequently references these equations using regular typeface $s$ on Page 5. Since $s$ (non-bold) has a different meaning in this work, it will be better if you could use the correct mathematical notation.
3) In the proof of Theorem 2.4, $\tilde{\Theta}_0$ should use \widetilde to ensure consistent notation throughout.
4) On Page 8, the paragraph beginning with "Limited-memory BFGS" is missing a period at the end.

**Strengths And Weaknesses:**

#Strengths
1) The experimental results are highly comprehensive. The paper conducts extensive evaluations on image classification, image reconstruction, and neural language modeling, which fully demonstrates the effectiveness of the ARCs-LSR1 algorithm.
2) The paper proposes the first quasi-Newton approach that is used in an adaptive regularized cubics setting.

#Weaknesses
1) The proposed ARCs-LSR1 algorithm lacks novelty, as its core components and analysis are both primarily derived from existing works, including Erway et al. (2020) and Cartis et al. (2011).
2) No implementation code is provided in this work.

---

### Note · Authors · 2025-07-12

**Comment:**

Hello Reviewers and editor,

  We thank all of you for your insightful comments and requests.

Given the timeline of 2 weeks and the requested changes, it would be rather difficult for us to accomplish all of these in that time span.

Thus, we will proceed with withdrawing the paper.

**Withdrawal Confirmation:**

I have read and agree with the venue's withdrawal policy on behalf of myself and my co-authors.